# Interface confined hydrogen evolution reaction in zero valent metal nanoparticles-intercalated molybdenum disulfide

Zhongxin Chen[1,2,*], Kai Leng[1,*], Xiaoxu Zhao[1,2], Souradip Malkhandi[1], Wei Tang[1,3], Bingbing Tian[1], Lei Dong[4], Lirong Zheng[5], Ming Lin[3], Boon Siang Yeo[1] & Kian Ping Loh[1]

Interface confined reactions, which can modulate the bonding of reactants with catalytic centres and influence the rate of the mass transport from bulk solution, have emerged as a viable strategy for achieving highly stable and selective catalysis. Here we demonstrate that 1T′-enriched lithiated molybdenum disulfide is a highly powerful reducing agent, which can be exploited for the *in-situ* reduction of metal ions within the inner planes of lithiated molybdenum disulfide to form a zero valent metal-intercalated molybdenum disulfide. The confinement of platinum nanoparticles within the molybdenum disulfide layered structure leads to enhanced hydrogen evolution reaction activity and stability compared to catalysts dispersed on carbon support. In particular, the inner platinum surface is accessible to charged species like proton and metal ions, while blocking poisoning by larger sized pollutants or neutral molecules. This points a way forward for using bulk intercalated compounds for energy related applications.

[1] Department of Chemistry and Centre for Advanced 2D Materials (CA2DM), National University of Singapore, 3 Science Drive 3, Singapore 117543, Singapore. [2] NUS Graduate School for Integrative Sciences and Engineering, National University of Singapore, Centre for Life Sciences, #05-01, 28 Medical Drive, Singapore 117456, Singapore. [3] Institute of Materials Research and Engineering, 2 FusionopolisWay, Singapore 138634, Singapore. [4] Department of Macromolecular Science, Fudan University, 220 Handan Road, Shanghai 200433, China. [5] Beijing Synchrotron Radiation Facility (BSRF), Institute of High Energy Physics, Chinese Academy of Sciences, Beijing 100049, China. * These authors contributed equally to this work. Correspondence and requests for materials should be addressed to K.P.L. (email: chmlohkp@nus.edu.sg).

Two-dimensional (2D) molybdenum disulfide (MoS₂) crystal exhibits high catalytic activity for hydrogen evolution reaction (HER), thus it is being considered as a low cost alternative to the commonly used platinum catalyst[1–5]. It is well known that the basal plane of MoS₂ is relatively inert and only the edge sites turn over reactions. MoS₂ can exist as the thermodynamically stable 2H phase (space group, P63/mmc) or the metastable 1T phase (space group, $P\bar{3}m1$). Bulk 2H-MoS₂ is a semiconductor with an indirect band gap of ∼1.2 eV, while the 1T-MoS₂ phase is metallic[4]. The 2H-to-1T′ MoS₂ phase conversion, induced commonly by chemical intercalation with alkali metal ions, has been applied as a strategy to enhance HER activities recently[1,6]. The increased electron density in Mo 4d orbitals due to electron transfer from intercalated lithium increases the metallic character of the basal plane and improves catalytic activity. However the 1T′ phase is metastable with respect to the 2H phase due to its intrinsic reactivity, this is further compounded by the rapid hydration of Li in aqueous phase, thus any prospect of long term, stable operation is quickly offset by the short-lived nature of these systems[3,7].

The use of 2D MoS₂ nanosheets as catalysts requires tedious exfoliation processes, which constrains mass production and industrial applications. Although, it is possible to coat metal nanoparticles on exfoliated 2H-MoS₂ to enhance its catalytic activity, such nanoparticles rapidly corrode under acidic HER environment and get leached, leading to the loss of activity[2,8]. Restacked MoS₂ nanosheets intercalated with transition metal ions can be synthesized from single layer MoS₂ dispersions by an ion exchange method, in which the cation (M²⁺) neutralizes the negative charge of the MoS₂ layer and the material restacks with alternating layers of MoS₂ and M(OH)₂. However, the chemical reduction of these metal ions requires the use of reducing agents and adds to the complexity of the process[9]. Alternatively, it is worth relooking at bulk MoS₂. Due to its layered structure, bulk MoS₂ can serve as host for a large class of metal nanoparticles within its inner planes and protect them from corrosion[9]. An intriguing idea is whether the host and intercalant can operate synergistically to achieve highly stable HER catalysis with the added benefit of a reduced loading of the noble catalyst, compared to coating on the bare, exposed surfaces of exfoliated 2D sheets.

A diffusion method relying on capillary action is primarily used to introduce nanoparticles into inner cavities of the host material. For example, the encapsulation of metal/metal oxide nanoparticles in carbon nanotubes, zeolites and metal organic frameworks produced enhanced catalytic performance towards alcohol oxidation[10], Fischer–Tropsch synthesis[11] and asymmetric hydrogenation[12]. However, encapsulation (or so-called intercalation) in 2D, layered materials is challenging due to the strong van der Waals force between adjacent nanosheets. The narrow spacing between the layers means that only small, alkali metal ions (Li⁺, Na⁺ or K⁺) can be electrochemically intercalated into bulk materials[13,14]. The insertion of larger cations like Co²⁺ usually requires the exfoliation of MoS₂ in water and their subsequent flocculation in the presence of the cation intercalant to generate a restacked, sandwiched structure.

Herein, we demonstrate a simple, straightforward strategy to achieve zero-valent intercalation in bulk MoS₂ by the in-situ reduction of noble metal ion precursors. Our method relies on the pre-intercalation of bulk MoS₂ with Li ions to form 1T′ phase-engineered LiₓMoS₂. Due to its enhanced electron density, we found that 1T′-LiₓMoS₂ has a large reduction potential, and this property can be exploited to reduce a large class of transition metal ions. The enlarged interlayer spacing in LiₓMoS₂ allows diffusional-ion exchange with the metal salt of interests, leading to the reduction of metal ion within the bulk of MoS₂ and removal of the intercalated Li⁺ as salt. The synergetic host–guest interaction in Pt-intercalated 2H MoS₂ (abbreviated as Pt-MoS₂) allows ultra-stable, long-term operation in HER with a reduced loading of Pt compared with commercial Pt/MoS₂ catalyst.

## Results

**Strong reducing power of 1T′-LiₓMoS₂.** 1T′-phase MoS₂ is unstable against 2H conversion in water, thus any long-term application in aqueous phase will involve the 2H phase inevitably. Our working hypothesis is this: the enhanced reactivity of the 1T′ phase compared with the 2H phase can be exploited to reduce metal ions and make nanometal-intercalated MoS₂ hybrids. To assess its electron reducing ability, the reduction of fullerene (C₆₀) is chosen as a model system[15]. C₆₀ is an excellent electron acceptor, which can undergo distinct six successive reductions in solution. The ∼0.5 V separation between each reduction step of the C₆₀ means that it is relatively easy to identify the reduction potential of a particular reducing agent[16]. In this regard, the chemical reduction, as well as the electrochemical reduction of C₆₀ by 1T′-LiₓMoS₂ is first studied. As shown in Fig. 1a,b, in argon-protected environment, C₆₀ can be readily reduced to C₆₀⁻ (fulleride) by LiₓMoS₂, producing an obvious colour change from purple to orange. The existence of fulleride is confirmed using ultraviolet–vis spectroscopy, where four fingerprint peaks located at 923, 995, 1,033 and 1,075 nm, arising from allowed $t_{1u} \rightarrow t_{1g}$ transitions and its vibrational fine structure, can be seen. The presence of fulleride also gives rise to a broad signal with a low g value (2,000) in the electron paramagnetic resonance (EPR) spectrum, which is due to the spin-orbit coupling effects of unquenched angular momentum in Jahn–Teller-distorted states of the $(t_{1u})^1$ configuration[15]. We further compare LiₓMoS₂ with a series of common reducing agents for the chemical reduction of C₆₀, as shown in Supplementary Fig. 1. Similar to strong reducing agents like hydrazine (−1.20 V versus RHE), sodium borohydride (−1.24 V versus RHE), 1T′-LiₓMoS₂, is able to spontaneously reduce C₆₀ in the first of the six-electron reduction reaction (−0.68 V versus RHE) at room temperature, while weak reducing agents and 2H-MoS₂ cannot.

The reduction ability of 1T′-phase LiₓMoS₂ is further examined by the electrochemical reduction of C₆₀ (ref. 17). C₆₀ is distinguished by a six-electron reduction process, which can be revealed by six distinct pairs of peaks in cyclic voltammetry and six peaks in differential pulse voltammetry in Supplementary Fig. 2. The modification of electrode with 1T′-phase LiₓMoS₂ results in a positive shifts (∼0.2 V) of these redox pairs together with a significant enhancement in the current density in Fig. 1c. The first redox pair (C₆₀⁻ fulleride) almost disappears due to the spontaneous pre-reduction of C₆₀ in the presence of LiₓMoS₂. Increasing the scan rate to 500 mV s⁻¹ did not affect the intensity of the reduction peaks, which indicates fast electron transfer kinetics and excellent conductivity of LiₓMoS₂ (Supplementary Fig. 3)[16]. In contrast, 2H-MoS₂ acts as a passive electrode in the electrochemical reduction of C₆₀, its peak positions are similar to that of glassy carbon in Supplementary Table 1. On the basis of this study, we can infer that 1T′-LiₓMoS₂ is a strong reducing agent with a reduction potential more negative than −0.7 V versus RHE.

**Ion exchange and intercalation of noble metals.** The strong reduction ability of 1T′-LiₓMoS₂ can be exploited for the synthesis of hybrid MoS₂ nanocomposites consisting of zero-valent metal intercalants. As shown in Supplementary Fig. 4, we first screen the first two rows of transition metal precursors, including PtCl₆²⁻ (1.48 V, versus RHE), AuCl₄⁻ (0.93 V), Pd²⁺ (0.92 V), Ag⁺ (0.80 V), Ru³⁺ (0.70 V), Cu²⁺ (0.34 V), Ni²⁺ (−0.25 V),

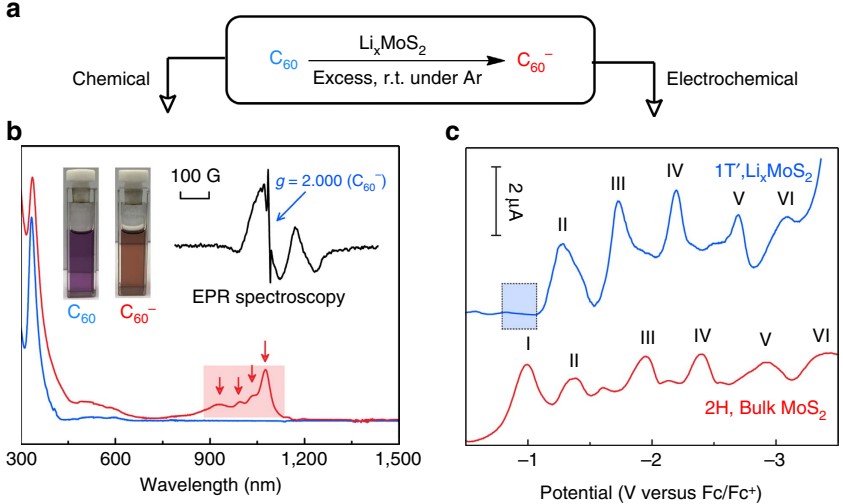

**Figure 1 | Strong reduction ability of 1T'-Li$_x$MoS$_2$ as evidenced by the spontaneous reduction of C$_{60}$.** (**a**) C$_{60}$ reaction equation. (**b**) UV–vis–NIR spectra and digital photos before/after Li$_x$MoS$_2$ reduction. Inset: solution EPR spectrum of the generated C$_{60}^-$ in AN/Toluene. (**c**) Differential pulse voltammetry curves of electrochemical reduction of C$_{60}$ on 1T'-Li$_x$MoS$_2$ and 2H-MoS$_2$ in AN/Toluene.

Co$^{2+}$ ($-0.28$ V) and Fe$^{2+}$ ($-0.44$ V), with their standard reduction potential indicated in the brackets. The reduction is performed in a non-aqueous solution (anhydrous THF or NMP depending on the solubility of precursors) to prevent the de-stabilization of the 1T'-phase[18]. In this anhydrous environment, the high-valent state metal ions diffuse into the layers of 1T'-phase Li$_x$MoS$_2$ and are *in-situ* reduced to the zero-valent state. To compensate for the excess charges, Li$^+$ and Cl$^-$ ions are completely leached from the MoS$_2$ host after this ion exchange. It must be pointed out that bulk 2H-MoS$_2$ and exfoliated MoS$_2$ nanosheets cannot reduce PtCl$_6^{2-}$ to Pt(0) (Supplementary Fig. 5); exfoliated MoS$_2$ nanosheets only have very weak reduction ability due to the rapid destruction of the 1T'-phase after hydration.

$$\text{Li}_x\text{MoS}_2 \xrightarrow{\text{PtCl}_6^{2-}} x\text{Li}^+ + x\text{Cl}^- + \text{Pt}(0)$$
$$+ \text{MoS}_2 \xrightarrow{\text{Exchange}} \text{PtMoS}_2 + x\,\text{Li}^+\text{Cl}^- \tag{1}$$

According to the equation above, Pt-MoS$_2$ is prepared from the intercalation of bulk MoS$_2$ with lithium using n-BuLi, followed by subsequent redox reaction with sodium hexachloroplatinate. As illustrated in Fig. 2a, bulk, 1T'-phase Li$_x$MoS$_2$ is directly used as reductant and host materials, which bypasses the tedious exfoliation process to form 2D and restacked MoS$_2$. More importantly, Pt nanoparticles are sandwiched between adjacent MoS$_2$ layers, thus protecting them against chemical corrosion and minimizing their dissolution and aggregation during long-term HER test[19].

To assay for the spatial distribution of intercalated and reduced zero-valent Pt, time-of-flight secondary ion mass spectroscopy (ToF-SIMS) is used to acquire the 3D elemental distribution of Pt in MoS$_2$ (ref. 20). As shown in Fig. 2b, Pt nanoparticles can be clearly seen on MoS$_2$. At a longer sputtering time, we observe both Pt$^-$ and MoS$_2^-$ signals in the composition map of the deeper layers, which proves the successful intercalation of Pt at deeper regions of MoS$_2$ single crystal ($\sim$200 nm). From the sputtering profile, Pt$^-$ species can be detected at depths up to 400 nm and beyond. The SIMS profiles of MoS$_2^-$ and Pt$^-$ signals in Fig. 2c suggest that the vertical distribution of Pt$^-$ is quite homogeneous. Consistent with the kinetics of the intercalation process, a diffusion gradient exists for the Pt nanometals when we

analysed its lateral distribution on layers exfoliated from bulk Pt-MoS$_2$ by the 'Scotch tape' method. Pt signal can be detected by EDS mapping only at the edge of MoS$_2$ after 12 h of intercalation, while the signal can be detected homogeneously across the flake after 3 days, as seen in the FESEM and EDS images in Supplementary Figs 7 and 8. Pt-MoS$_2$ has a well-ordered layered structure with an expanded layer spacing due to the intercalation of the metal nanoparticles[18]. Its bulk morphology is markedly different from the restacked nanosheets generated from exfoliation strategy. The selected area electron diffraction (SAED) pattern of Pt-MoS$_2$ is distorted from the typical hexagonal pattern of bulk 2H-MoS$_2$. The continuous ring with a measured d-spacing of 2.2 Å as determined by Fourier transform can be assigned to the {111} planes of Pt, while the appearance of two new rings strongly suggests the intercalation of Pt species into MoS$_2$ host structure (Supplementary Fig. 9)[8]. The intercalation strategy was also verified to be successful for a wide range of transition metals, such as Au, Ru and Pd as verified by SEM and EDX (Supplementary Figs 7 and 8).

The change in interlayer spacing of MoS$_2$ after intercalation by Pt was also verified by grazing incidence X-ray diffraction (GIXRD). As shown in Fig. 3, we can find a broad peak at $\sim$2.46° ($\sim$3.6 nm) in 1D and 2D GIXRD for the MoS$_2$|Pt|MoS$_2$ sandwiched structure, which is absent in the bulk sample. The d-spacing of MoS$_2$ layer is calculated to be 0.63 nm from X-ray diffraction and the average size of Pt nanoparticles is found to be $\sim$2 nm from HRTEM images in Supplementary Fig. 6. Accordingly, Pt-MoS$_2$ has a repeat unit of 2–3 layers of MoS$_2$ and 1 layer of Pt nanoparticles. In addition, we also observe a peak shift of the MoS$_2$ {002} from 14.10 to 14.34° together with an increase in FWHM after Pt intercalation, which can be ascribed to the stress on the MoS$_2$ layers induced by Pt nanoparticles in-between[21,22]. The existence of Pt nanoparticles in the inner layers of MoS$_2$ is directly confirmed by the TEM images of exfoliated Pt-MoS$_2$. As shown in Fig. 3d,e, Pt nanoparticles with sizes ranging from 1.5 to 3.5 nm are homogeneously dispersed on the MoS$_2$ flakes. The SAED pattern (Fig. 3f, as indicated by red arrow) and EDS spectrum (Fig. 3g) confirm the successful intercalation of Pt nanoparticles into the inner layers of MoS$_2$ flakes. The cross-section HAADF-STEM images, as well as the EELS mapping in Supplementary Fig. 6 also demonstrate the intercalation of Pt nanoparticles in between the MoS$_2$ layers. The

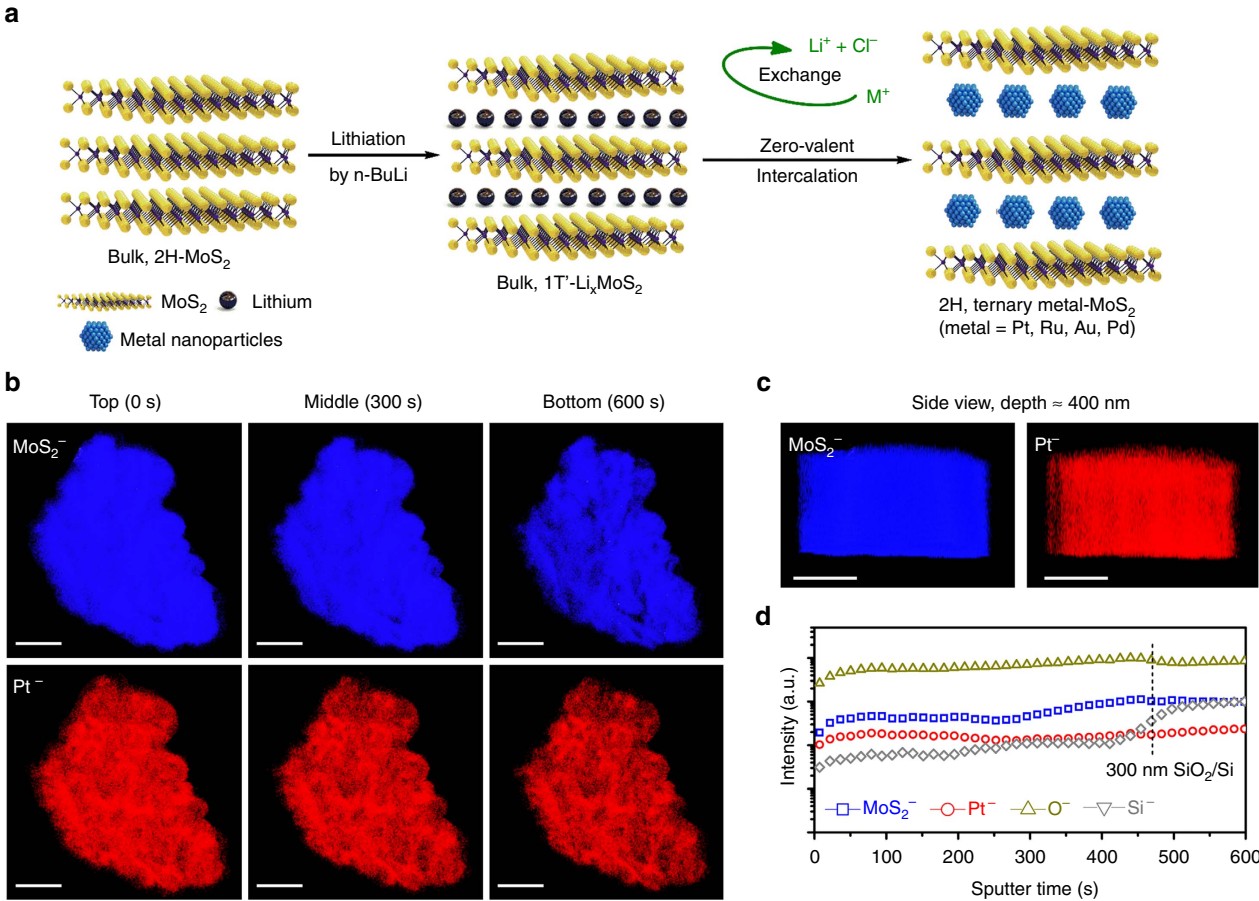

**Figure 2 | Homogeneous metal distribution in bulk MoS$_2$.** (**a**) Zero-valent intercalation of metal nanoparticles by an *in-situ* reduction strategy. (**b**) ToF-SIMS maps at the beginning (0 s), middle (300 s) and end (600 s) of the experiment. The false colours of blue and red corresponds to MoS$_2^-$ (162 amu) and Pt$^-$ (195 amu). (**c**) ToF-SIMS side view and (**d**) depth profile of the same region (etching depth is roughly 400 nm, as calculated by the step of 300 nm SiO$_2$/Si wafer). Scale bar, 20 μm.

small-angle X-ray scattering (SAXS) profiles in Supplementary Fig. 12 confirm a loosely stacked structure of Pt-MoS$_2$ due to hydrogen bubbles-induced volume expansion during zero-valent intercalation (Supplementary Note 1).

**Structure evolution of intercalated compounds.** The phase conversion of 2H and 1T′ during *in-situ* reduction is followed by X-ray photoelectron spectroscopy (XPS). As shown in Fig. 4a and Supplementary Fig. 10, chemical lithiation of bulk MoS$_2$ converts it to the metallic 1T′-phase with the 1T′-related peaks at 228 and 231 eV for the spin orbit coupled Mo$_{3d}$ peaks, and 161.2 and 162.6 eV for the S$_{2p}$ peaks[18]. The Pt-intercalated MoS$_2$ reverts to the 2H-phase after electrons are donated to Pt ion precursor and the exchange-diffusion of the Li$^+$ ions. The intercalated Pt atoms are mostly at their zero-valence state as judged from the Pt$_{4f}$ XPS spectrum in Fig. 4b. The absence of Pt$^{4+}$ species suggests the full reduction of sodium hexachloroplatinate by Li$_x$MoS$_2$. The phase conversion from 1T′-Li$_x$MoS$_2$ to 2H-Pt-MoS$_2$ is also confirmed by the Raman spectra in Supplementary Fig. 11. The $J_1$ to $J_3$ phonon modes around 170–300 cm$^{-1}$ of the 1T′-phase MoS$_2$ are clearly discernible in the lithiated and exfoliated materials, while absent in Pt-intercalated 2H-MoS$_2$. The latter shows two main phonon peaks at 404 cm$^{-1}$ and 380 cm$^{-1}$, which are due to the A$_{1g}$ and E$^1_{2g}$ modes of the 2H phase[1,18].

To verify and quantify the diffusional substitution of Li ions by Pt ions, the elemental composition of the composite was analysed

by XPS (Supplementary Fig. 10) and ICP-OES (Supplementary Table 2). The residual lithium and chlorine percentage is < 0.5 wt% in Pt-MoS$_2$ and there are no Li$_{1s}$ or Cl$_{2p}$ peaks according to XPS analysis, which proves that most of the Li ions have been replaced by Pt. We also confirmed > 80% lithium exchange using the Mohr titration of chloride[23], where Li$^+$ is assumed as the only counter ion of Cl$^-$ in the supernatant of Pt-MoS$_2$ (Supplementary Fig. 13 and Supplementary Note 2).

The local bonding environment in Pt-MoS$_2$ was investigated by extended X-ray absorption fine structure (EXAFS)[7]. The K-edge EXAFS spectrum for pristine MoS$_2$, as shown in Fig. 4c, is consistent with the trigonal prismatic structure, with six Mo–S bonds of 1.93 Å and six Mo–Mo bonds of 2.85 Å. These peaks are much weaker in the case of Li$_x$MoS$_2$, which means the second shell of Mo atoms is disturbed; the decreased Mo–Mo distance of 2.67 Å also suggests the change in site symmetry due to trigonal prismatic (2H) to octahedral (1T′) phase change[24]. In the case of Pt-MoS$_2$, the K-edge EXAFS indicate structural rearrangement consistent with the 2H phase. We do not find evidence of Mo–Pt interaction. This suggests that the Pt atoms in MoS$_2$ are probably existing as Pt nanoclusters, as opposed to Pt occupying vacancy sites[7]. Powder X-ray diffraction was also performed to examine the layered structure of MoS$_2$. As presented in Fig. 4d, the strong {002} diffraction peak at 14.8° of bulk MoS$_2$ shifts to lower position after Li intercalation (14.3°), which can be ascribed to the {001} of 1T′-phase of Li$_x$MoS$_2$ (ref. 18). The intensity of this peak is reduced after Pt intercalation. The presence of

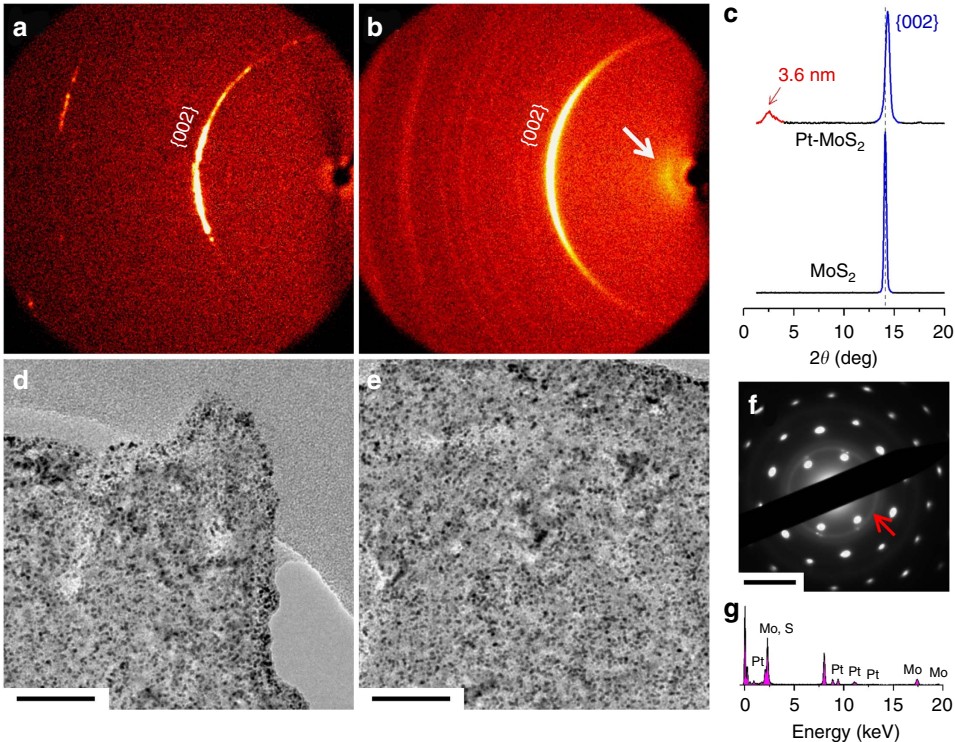

**Figure 3 | Evidence of Zero-Valent Intercalation.** (**a**) GIXRD patterns of single crystal $MoS_2$ and (**b**) Pt-$MoS_2$; (**c**) Corresponding 1D spectra in the out-of-plane direction. (**d**,**e**) TEM images of the nanosheets exfoliated from bulk Pt-$MoS_2$. Corresponding (**f**) SAED and (**g**) EDS spectrum of the exfoliated Pt-$MoS_2$. Scale bar, **d**,**e** 50 nm; **f** 5 nm$^{-1}$.

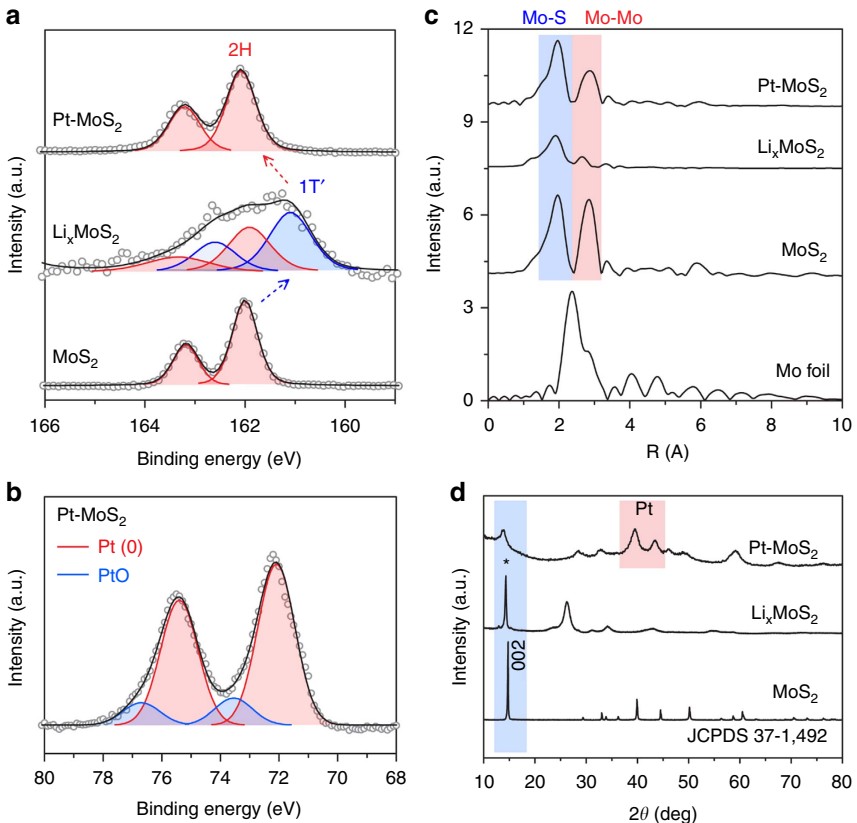

**Figure 4 | Phase transition process of metal intercalation.** (**a**) XPS $S_{2p}$ spectra showing the evolution from 2H (bulk $MoS_2$) to 1T' ($Li_xMoS_2$) to 2H (Pt-$MoS_2$); (**b**) XPS $Pt_{4f}$ spectrum validating the zero-valent state of intercalated metals; (**c**) $k^2$-weighted Mo K-edge EXAFS spectra of Pt–$MoS_2$ in comparison to Mo foil, bulk $MoS_2$ and $Li_xMoS_2$; and (**d**) XRD patterns showing the fcc structure of intercalated Pt nanoparticles.

Pt nanoparticles can be judged clearly from XRD peaks originating from the face-centered cubic (fcc) crystal phase of Pt, with two strong peaks at 39.4° {111}, 43.5° {200} and a weak peak at 67.5° {220}, respectively[8]. The peaks located at 32.8° and 52.1° can be assigned to the {100} and {110} of $MoS_2$.

**HER performance in a confined space**. To examine interface-confined catalytic reactions in the Pt-$MoS_2$ hybrid, in which the catalysts are intercalated within the quasi-2D $MoS_2$ layers, we chose HER as the model reaction because of the small hydrated radius and fast diffusion kinetics of proton in water, which has less mass diffusion limitation.

We measured the electrochemical HER performance using a three-electrodes cell and compared the performance with 40 wt% commercial Pt/C catalyst and exfoliated $MoS_2$ nanosheets (Supplementary Figs 14 and 15). The content of Pt, Ru, Pd and Au in $MoS_2$ are identified by ICP-OES as 10.4, 11.3, 24.5 and 18.3 wt%, respectively. Figure 5a shows that these metal intercalated $MoS_2$ are highly active toward HER. For instance, Pt-$MoS_2$ exhibits a negligible overpotential at current density of 10 mA cm$^{-2}$, together with a very low charge transfer resistance ($R_{CT}$) of < 5 Ω in the electrochemical impedance spectroscopy (EIS) spectra (Fig. 5b). The facile kinetics of Pt-$MoS_2$ is a major reason for its superior catalytic activity in HER (Supplementary Note 4)[5,25,26]. The $R_{CT}$ is much larger than the porosity resistance of the electrode ($R_P$) and is overpotential-dependent in Supplementary Fig. 16, suggesting the domination of $R_{CT}$ and a combination of Volmer–Tafel mechanism with the recombination of adsorbed $H^*$ as the rate determining step in HER reaction[27,28]. The growth of Pt within the confined layers of $MoS_2$ may lead to a smaller particle size, leading to the excellent HER performance. Importantly, it is observed that the long-term stability of these metal intercalated $MoS_2$ far exceeds that of commercial Pt/C in Fig. 5c. For instance, there is no obvious decay in overpotential for Pt-$MoS_2$ after > 30 h continuous

operation at 50 mA cm$^{-2}$, whereas commercial Pt/C is rapidly degraded during stability test as a result of Pt nanoparticles dissolution/aggregation, as well as carbon support corrosion (Supplementary Note 3)[19]. The sandwiching of Pt between the planes of $MoS_2$ has a synergetic effect in enhancing the stability and kinetics of the hydrogen evolution reaction (HER)[29]. This is a key advantage compared with the conventional way of coating metal catalysts on the surface of nanosheets, in which these metal catalysts are invariably corroded in acidic environment and leached away, contributing to catalyst failure and instability of HER performance[14]. In contrast, our Pt intercalated-$MoS_2$ system shows superior stability because of the synergetic host-intercalant effect (Supplementary Note 5). In addition, pressure-releasing channels in the form of cracks, voids and edges exist throughout the layered Pt-$MoS_2$ structure and these prevent pressure build-up during hydrogen evolution (Supplementary Figs 18 and 19).

We adopt two common indicators here to further access the HER performance. The Tafel slope is the increase in overpotential required to produce a one-order magnitude rise in current density while the onset potential is the potential at which current density begins to fall steeply due to proton reduction[30,31]. We select a similar overpotential range (25–45 mV) to calculate the Tafel slope for better comparison. As a benchmark catalyst, 40 wt% Pt/C shows a small Tafel slope of 22 mV dec$^{-1}$ and an onset potential of 24 mV for hydrogen evolution. The Tafel slope and onset potential for Pt, Pd, Ru, Au-intercalated $MoS_2$ are 25, 30, 61, 38 mV dec$^{-1}$ and 20, 35, 38, 22 mV in Fig. 5d, respectively. The mass transport limitation is not significant due to the fast diffusion kinetics of protons. The HER performance of various metal-intercalated $MoS_2$ follows the theoretical prediction for noble metals (so-called Volcano shape) except for Au-$MoS_2$ (ref. 5), suggesting that the active sites are on noble metal nanoparticles and $MoS_2$ served as co-catalysts and support for noble metals. The hydrated Au precursor has an excellent solubility in THF, leading to a much better intercalation

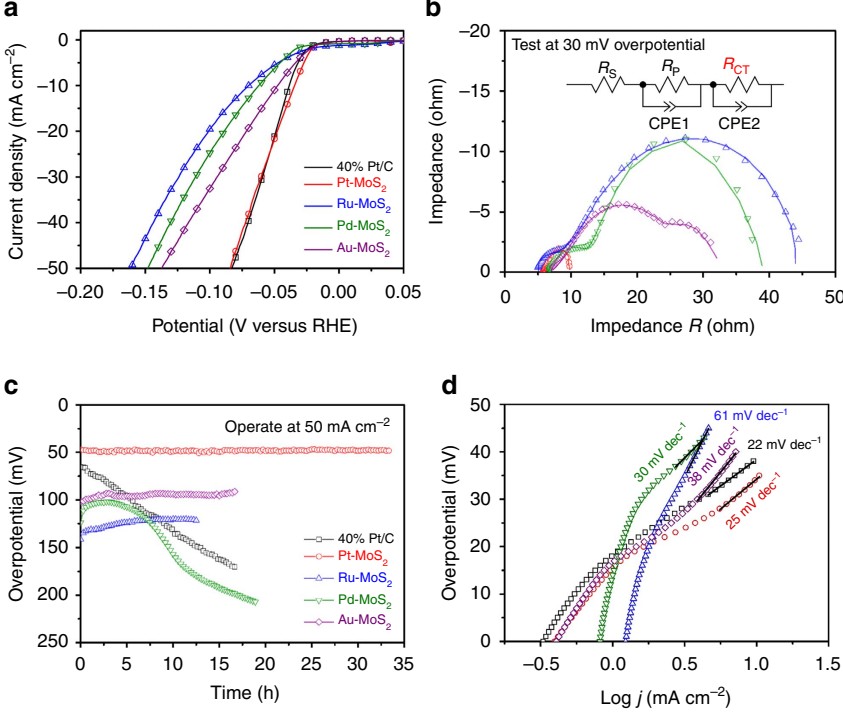

**Figure 5 | Superior HER performance of metal-intercalated $MoS_2$ catalysts.** (**a**) LSV curves, (**b**) Nyquist plots and EIS simulation model, (**c**) long-term stability test and (**d**) Tafel plots. All experiment were conducted in 0.5 M $H_2SO_4$ at room temperature.

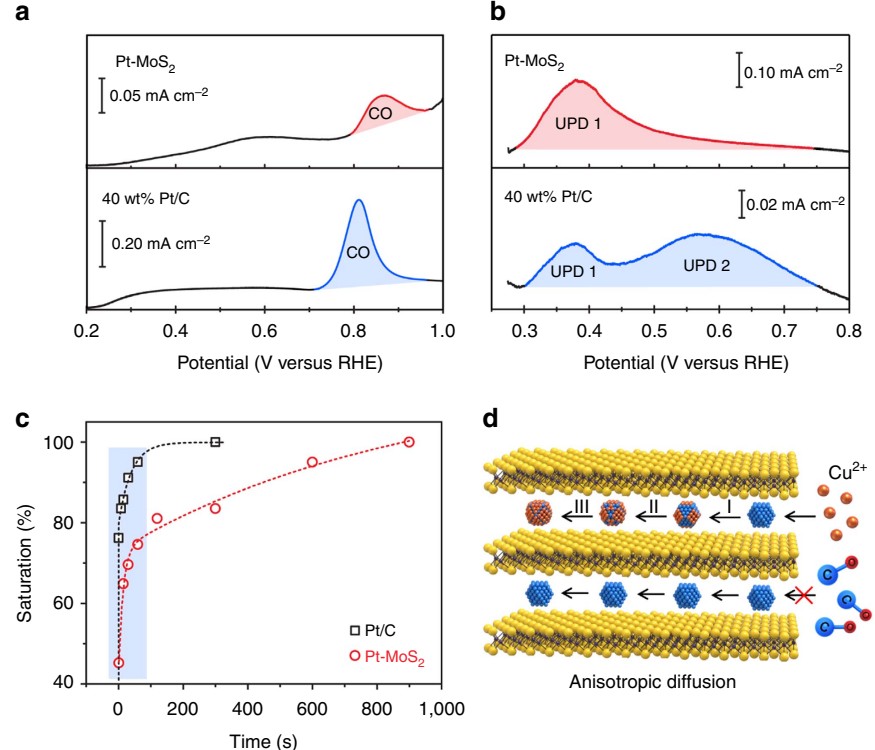

**Figure 6 | Catalysis mechanism of inner Pt nanoparticles.** (**a**) CO stripping and (**b**) Cu UPD of 40 wt% Pt/C and Pt-MoS$_2$, showing the different accessibility of inner Pt nanoparticles. (**c**) The saturation rate of the Cu monolayer deposition upon holding at a fixed UPD potential, which indicates a much slower diffusion process in the case of Pt-MoS$_2$ compared with Pt/C. (**d**) Schematics showing the anisotropic diffusion of CO gas and cupric ions into Pt-MoS$_2$.

efficiency compared to Ru and Pd, which may be a possible reason for the enhanced activity of Au-MoS$_2$. The similar trend in long-term stability also proves the validity of this zero-valent intercalation, while the relatively poor stability of Pd-MoS$_2$ may be attributed to its surface oxidation and dissolution of its oxide in the acidic environment. Furthermore, the performance of Pt-MoS$_2$ is compared with recently reported HER catalysts in Supplementary Table 4, showing its superior performance. The advantage of zero-valent intercalation of MoS$_2$ is that bulk MoS$_2$ powder is processed directly to make the hybrid, bypassing the tedious exfoliation process of 2D MoS$_2$ nanosheets. Thus we were able to demonstrate industrial scalability by fabricating a $5 \times 5$ cm$^2$ catalyst-loaded water splitting membrane in Supplementary Fig. 17. This is unprecedented in terms of research in 2D transition metal chalcogenide so far, which are often limited by the size of the exfoliated flakes and difficulty in spin-coating a continuous films[27].

One reason for the extra stability of Pt-MoS$_2$ arises from the passivation effect enjoyed by the Pt nanoparticles sandwiched between the MoS$_2$ layers. However, it is not known if the highly anisotropic diffusion needed to access the Pt in the inner layers will result in a lower catalytic efficiency. To evaluate the electrochemical activity of these MoS$_2$-sandwiched Pt catalysts, we carry out carbon monoxide (CO) voltammetric stripping and Cu underpotential deposition (UPD) (ref. 32). In stripping voltammetry, a saturated layer of CO is pre-adsorbed on the Pt catalyst followed by its subsequent oxidation in voltammetric scans. In UPD, Cu metal is deposited above the Nernst potential to form sub-to-monolayer Cu films on Pt surface (Supplementary Note 6). The presence of the CO stripping peak or UPD peak allows us to assay the electrochemical active surface area (ECSA) of the Pt nanoparticles (Supplementary Figs 20 and 21)[33]. In addition, kinetic information can be obtained by observing the

saturation rate of the Cu monolayer deposition when implemented in the chronoamperometric mode (see Supporting information for detailed discussion on UPD and CO stripping). As presented in Fig. 6a,b, the benchmark Pt/C shows comparable ECSAs in CO stripping (0.62 cm$^2$) and Cu UPD (0.58 cm$^2$). The CO stripping peak at 0.81 V versus RHE arises primarily from its saturated adsorption on Pt {111} (ref. 34), while the two broad Cu UPD peaks correspond to two distinct redox reactions: an intermediate copper layer coadsorbed with sulfate ions (Cu UPD1), as well as the $1 \times 1$ monolayer deposition (Cu UPD2) (ref. 35). However, the CO stripping peak is much weaker in the case of Pt-MoS$_2$. Due to the nanosized Pt particles in the MoS$_2$ confined layers, we only observe one strong Cu UPD peak in Pt-MoS$_2$, which is due to sulfate-coadsorbed copper layer. The ECSA as determined by CO stripping for Pt-MoS$_2$ (0.16 cm$^2$) is also much lower than that of Cu UPD (0.85 cm$^2$), implying that the Pt sites in Pt-MoS$_2$ are not accessible to CO adsorption, although these are accessible to cupric ions. The mass transport of Cu ions into the inner active surface of Pt-MoS$_2$ can be electrically driven; however neutral CO molecules rely entirely on mass diffusion and the anisotropic diffusion path involving through the open edges of the MoS$_2$ layers becomes rate-limiting (Supplementary Table 3).

Diffusion kinetics can be used to distinguish exposed Pt and trapped Pt in the electrode structure, it is investigated by holding the Cu UPD at a fixed voltage (275 mV versus RHE). Fig. 6c and Supplementary Figs 22–24 show that the current signal is rapidly saturated for Pt/C, whereas it takes a longer time to reach saturation for Pt/MoS$_2$ due to the slower diffusion to reach the sandwiched Pt nanoparticles. Such intercalated structure is promising for electrocatalysis because the inner surface is accessible to charged particles like proton and metal ions, while blocking poisoning by larger sized pollutants or neutral molecules

(Fig. 6d). Slow $O_2$ or CO diffusion is beneficial for long-term operation due to reduced corrosion or poisoning. Thus, the bulk, layered structure of Pt-$MoS_2$ imparts more stability than its randomly restacked counterparts, contributing to excellent activity and stability in HER.

## Discussion

The direct intercalation of zero-valent metals in bulk, layered materials has rarely been achieved. Previous methods rely on the exfoliation of 2D sheets from bulk crystals, followed by ion-exchange of the Li with metal salts in solution and the re-stacking of the sheets, where upon the metal ions got 'intercalated'[9]. Reduction to zero-valent metals requires additional chemical reduction steps, especially in the case of 2H-$MoS_2$, which does not have inherently strong reducing power. As demonstrated in this work, the *in-situ* reduction of Pt precursors by making use of the highly reducing 1T′-$Li_xMoS_2$ phase provides a convenient way of generating a zero valent-metal intercalated quasi-2D material. It involves the following steps: diffusion of Pt ions from solution into the inner planes of lithiated $MoS_2$, and reduction therein of the Pt ions by the metallic 1T′ phase, followed by Pt nanoparticle growth under layer-confined condition. At the same time, $Li^+/Cl^-$ ions will be ion exchanged from $MoS_2$, while the latter undergoes phase conversion from the 1T′-phase to 2H-phase. The metal ion exchange proceeds readily when hydrated metal salts like $Na_2PtCl_6 \cdot 6H_2O$ were used, since the large hydration energies of $Li^+$ favors its de-intercalation from $MoS_2$ (ref. 36). In contrast, when anhydrous metal precursors (for example, $K_2PtCl_4$) were used, it was observed that large nanoparticles nucleate at the edge or surface of $MoS_2$ flakes due to its sluggish diffusion. The negative enthalpy of Pt reduction by 1T′ phase ($-571 \, kJ \, mol^{-1}$)[37], along with the energy gain from phase conversion from 1T′-phase to 2H phase ($> -68 \, kJ \, mol^{-1}$ from reduction potential) and the large solvation energy gain of $Li^+$ ($-101 \, kJ \, mol^{-1}$ in THF)[38], contribute to the overall free energy gain of the ion exchange process.

The confined space within the adjacent $MoS_2$ layers provides an intriguing environment for catalysis[15,39,40]. First, the narrow space limits the growth of the Pt nanoparticles, and dispersion within the $MoS_2$ layers prevents aggregation. Thus even at a reduced weight loading (10 wt%) of Pt in Pt-$MoS_2$, it shows a higher ECSA value compared to benchmark (40 wt%) Pt/C. This is consistent with the larger electrochemical surface area of nanoparticles grown which are size-confined. For instance, Cameron *et al.*[10] reported that zeolite-confined $RuO_2$ nanoparticles have a very small particle size ($1.3 \pm 0.2 \, nm$). Importantly, such spatial restriction can also modify the accessibility and interaction between catalysts and reactants. Bao *et al.*[41] observed strong deformations of transition metals within the CNT channels due to different electronic structures and spatial confinement. This will lead to downshifted d-band states and weaker adsorption of CO, $N_2$ and $O_2$ molecules, which could also apply to Pt-$MoS_2$. Due to the restricted mass diffusion of CO, Pt-$MoS_2$ possess CO-tolerant properties, thus it can be potentially applied in methanol oxidation fuel cell where the CO poisoning of Pt catalyst is a serious problem.

In conclusion, we have demonstrated that 1T′ polymorph-enriched bulk $Li_xMoS_2$ is a powerful reducing agent. As evidenced by the chemical and electrochemical reduction of $C_{60}$ molecules, the reduction potential of 1T′-phase $Li_xMoS_2$ is more negative than $-0.7 \, V$ versus RHE. The reducing power can be exploited to fabricate zero-valent metal intercalated 2H-$MoS_2$ for a wide class of transition metals. Taking advantage of the enlarged interlayer spacing in 1T′-phase $Li_xMoS_2$, the diffusional exchange of noble metals ions, followed by its *in-situ* reduction, can occur

within the layers of $MoS_2$, giving rise to a unique hybrid quasi-2D system, that is, noble-metal intercalated $MoS_2$. Although, there are plenty of reports on the use of 2D $MoS_2$ in HER, 1T′-phase $MoS_2$ is not stable for long-term practical applications while the performance of 2H-phase is not satisfactory and require defect engineering. The advantages of Pt-intercalated 2H-$MoS_2$ include the size-restricted growth of Pt nanoparticles within $MoS_2$, where the Pt remains highly active for HER, and the slow kinetics for poisoning species in confined space, which ensures that the long-term stability of this hybrid system exceeds that of commercial Pt/C catalyst. This study points to the possibility of a large class of zero valent metal-intercalated TMDs exhibiting synergetic properties in catalysis.

## Methods

**Synthesis of Pt-$MoS_2$.** Around 1 g of $MoS_2$ powder was added to 20 ml 1.6M n-BuLi in hexane, and stirred at r.t. for 2 days in an argon-filled glove box. Excess amount of n-BuLi was removed by centrifugation. Then, 100 mg $Li_xMoS_2$ was dispersed in 20 ml anhydrous THF and 33.7 mg $Na_2PtCl_6 \cdot 6H_2O$ was added. The mixture was sealed in a Teflon-lined autoclave and kept at 80 °C for 2 days. As-prepared Pt-$MoS_2$ was thoroughly washed with THF, IPA, ethanol and water (Supplementary Methods).

**Equipment.** The following equipment were used: Raman (WITec Alpha 300R), SEM/EDS (Jeol JSM-6701F), AFM (Dimension Fast Scan), XPS (AXIS UltraDLD, monochromatic Al $K_\alpha$), Electrochemistry (CHI 660E and Zahner Zennium with a three-electrodes cell, Supplementary Methods), STEM & EELS (Nion UltraSTEM-100 with aberration-correction, 60 KV), TEM/EDS (FEI Titan, 80 kV), ToF-SIMS (ION-TOF SIMS5 with $Bi^+$ and $Cs^+$ beams, Supplementary Methods), EXAFS (1W1B-XAFS beamline, BASF, Supplementary Methods), SAXS (SAXSess mc2), XRD/GIXRD (Bruker D8 & GADDS, Supplementary Methods).

**Data availability.** The authors declare that the data supporting the findings of this study are available within this paper and its Supplementary information file, or from the corresponding authors.

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

## Acknowledgements

We thank National Research Foundation, Prime's Minister Office for support under the mid-sized research centre (CA2DM). We acknowledge Ms Zilu Niu and WinTech Nano-Technology for ToF-SIMS measurement, Dr Yonghua Du for sample preparation in the EXAFS experiment, Ms Qianqian Hu for her assistance with the XPS, Ms Guangrong Zhou for TEM cross-section sample preparation, Dr. Ping Yang for 1D GIXRD measurements and Prof Richard D. Webster, Dr Zheng Long Lim, Mr Hejian Zhang and Dr Guo-hong Ning for their contributions on EPR. Z.C. thanks the NGS Scholarship for support. K.L. appreciates scholarship support by Solar Energy Research Institute of Singapore (SERIS).

## Author contributions

Z.C., K.L. and K.P.L. conceived the research and wrote the draft. Z.C. and K.L. synthesized the materials and performed the electrochemical measurements. Z.C. and X.Z. conducted TEM characterization and data analysis. S.M. and B.S.Y. performed the CO and Cu tests. W.T. and B.T. assisted in the materials characterization and data analysis. L.Z. performed the EXAFS measurement, M.L. performed the GIXRD measurement and L.D. acquired the SAXS patterns. All authors discussed and commented on this manuscript.

## Additional information

**Competing financial interests:** The authors declare no competing financial interests.

