## [Peer Review File · Nature Communications]

Reviewers' Comments:

Reviewer #1 (Remarks to the Author)

This manuscript reports a new finding of intercalating and stabilizing noble metal particles between the layers of bulk MoS₂. Those noble metal intercalated MoS₂ shows good HER activity and stability. Also, those intercalated metal particles are accessible only for charged species, not neutral molecules. The results are generally interesting for publication. There are some questions that need to be addressed.

1. Fig. 2B, the manuscript says "A diffusion gradient is apparent from the lateral distribution map". It seems to suggest that the concentration of Pt is high near the peripheral regions and low in the center due to the diffusion process. However, Fig. 2B does not seem to be the case by naked eye. It appears that there are random locations of high Pt concentration throughout. The authors should explain why if this is true. In addition, it is more convincing to do a cross-section TEM to confirm the intercalation of Pt between MoS₂ and show the average size of Pt particles.

2. Fig. 4a, why does the HER activity rank in the order of Pt > Au > Pd > Ru? If the role of MoS₂ is a co-catalyst and support, the HER activity would follow the intrinsic activity of the noble metals. The Au should have the lower activity than Pd at least.

3. Fig. 4c, why is the only Pt-MoS₂ showing good stability, not others? Based on the rationale presented here, MoS₂ should stabilize all the intercalated metals.

4. Fig. 4d, for the Tafel plots, the authors need to explain why they only use the top part to calculate the slopes.

Reviewer #2 (Remarks to the Author)

The intercalation of layered MoS₂ with other species is well-documented. It is also well-established that Li_xMoS₂ is a strong reducing agent capable of chemically reducing H₂O to produce H₂ leading to exfoliated MoS₂. So it is not surprising that 1T'-Li_xMoS₂ can reduce transition metal ions, as claimed in this work. Overall, this work lacks the originality and novelty for a high-profile journal like Nature Communications.

More issues should be addressed:

1. Since 2H-MoS₂ is a semiconductor, bulk 2H-MoS₂ should suffer from limited electron transfer. The Pt-MoS₂ shows a low charge transfer resistance which could be attributed to Pt-enhanced conductivity. However, TEM data (Fig. 8) only reveal the formation of discrete Pt nanoparticles.

2. Solid evidence is absent to confirm that all Pt nanoparticles are exclusively intercalated into MoS₂.

3. The authors argued that the inner surface is accessible to protons for hydrogen evolution. If it is the case, the high pressure with the accumulated hydrogen bubble in the inner space is believed to drive the exfoliation of bulk MoS₂. It is supposed that only Pt species at the edge contributes to the HER activity.

4. The use of Pt wire as the counter is also problematic because trace Pt from anode dissolution can enhance the HER activity of the cathodic catalyst.

Reviewer #3 (Remarks to the Author)

The manuscript contributed by Chen et al. reports a method for directly loading noble metal nanoparticles on 1T-MoS₂ through a so-called in-situ reduction of noble metal precursor ions with reductive 1T-MoS₂. The main conclusion the authors would like to deliver is that the small metal nanoparticles (e.g., Pt nanocrystals) are formed in the interlayer gaps of the MoS₂ crystals. However, the conclusion is NOT supported with any direct and convincing evidence even though various characterization techniques have been used to characterize the samples.

If the Pt nanoparticles with several nanoparticles are sandwiched in the interlayer gaps of MoS₂, the interlayer spacing of the MoS₂ containing Pt nanoparticles has to be on the length scale.

However the XRD pattern of the MoS₂ with Pt nanoparticles still show the interlayer spacing of MoS₂ is similar to that of 2H bulk MoS₂ and 1T-LixMoS₂ (Fig. 3D showing their XRD peaks at similar positions). This is the first reason that I do not believe that the Pt nanoparticles are confined in small interlayer gaps. The Pt nanoparticles are more likely deposited on the outside of basal surfaces of MoS₂ powders. Otherwise, how can the 6.2-Å gaps accommodate the Pt nanoparticles with several nanoparticles?

If the Pt nanoparticles are confined in the interlayer gaps of MoS₂, all the single-layer MoS₂ layers should be covered with Pt nanoparticles if the Pt-modified MoS₂ is exfoliated. Can the authors provide the results?

The ToF-SIMS measurements shown in Fig. 2 canNOT support the conclusion of confining Pt nanoparticles in the interlayer gaps of MoS₂. The scale bars shown in Fig. 2 represent 20 microns, several orders (3-4) larger than the size of Pt nanoparticles as well as the interlayer spacing of MoS₂. How can the authors conclude that the Pt nanoparticles with several nanoparticles are confined in the interlayer gaps of MoS₂. The spatial resolution of the ToF-SIMS mapping cannot support the claim on several-nanometer Pt nanoparticles.

It is questionable of the results shown in Fig. S11 (Mohr Titration of chloride ions). The chloride ions existing in reaction solution can be significantly contributed by the Pt precursor ions (i.e., PtCl₆²⁻). As a result, the measured concentration of Cl⁻ is not solely determined by the exchanged Cl⁻ from the 1T-LixMoS₂ sample. Therefore, the conclusion derived from this measurement is not reasonable.

Overall, the results are still premature to reach a convincing conclusion although many characterizations have been done. It is suggested to claim the conclusion that can be derived from the real observations rather than claim a fancy conceptual conclusion without enough/convincing support. It is not ready for being published until enough and convincing evidences are provided to support the confinement of Pt nanoparticles in the narrow interlayer gaps of MoS₂.

Response to Referees:

Reviewer #1:

We thank the Reviewer for his/her positive assessment. To answer his concern:

The results are general interesting for publication. There are some questions that need to be addressed.

1. Fig.2B, the manuscript says "A diffusion gradient is apparent from the lateral distribution map". It seems to suggest that the concentration of Pt is high near the peripheral regions and low in the center due to the diffusion process. However, Fig. 2B does not seem to be case by naked eye. It appears that there are random locations of high Pt concentration throughout. The authors should explain why if this is true.

ANS: Thank you for the comment. We tested a fully intercalated Pt-MoS₂ sample for SIMS mapping so the diffusion gradient in Figure 2B is not so obvious. **Kinetic study** on the Pt diffusion has been carried out to observe this diffusion gradient. We performed Pt intercalation using single-crystal MoS₂ flakes and collected samples at 12 and 72 hours. The surface layers of Pt-MoS₂ were removed by "Scotch tape" method to expose the inner layers. We found that Pt nanoparticles can be only seen at the edge of MoS₂ after 12 hours intercalation in SEM image and EDS mapping, while it was fully intercalated after 3 days. This suggests that it is a diffusion process.

In addition, it is more convincing to do a cross-section TEM to confirm the intercalation of Pt between MoS₂ and show the average size of Pt particles.

ANS: We have supplied **cross-section HRTEM** (Fig. 1, see response of Reviewer #2, question 3) in the revised version of the paper, where the average size of Pt particles is 2 nm.

2. Fig. 4a, why does the HER activity rank in the order of Pt>Au>Pd>Ru? If the role of MoS₂ is a co-catalyst and support, the HER activity would follow the intrinsic activity of the noble metals. The Au should has the lower activity that Pd at lease.

ANS: We thank the referee for his/her comment. The hydrated Au precursor has an excellent solubility in THF, leading to a much better intercalation efficiency compared to Ru and Pd. Since the difference in HER activity of Au-MoS₂ and Pd-MoS₂ is not very big, this may be a possible reason for the enhanced activity of Au-MoS₂.

3. Fig. 4c, why is the only Pt-MoS₂ show good stability, not others? Based on the rational presented here, MoS₂ should stabilize all the intercalated metals.

ANS: We thank the referee for his/her comment. As presented in Figure 4c in main text, Au-MoS₂ and Ru-MoS₂ also show enhanced stability compared to benchmark Pt/C. The poor stability of Pd-MoS₂ is attributed to the oxidation of Pd particles in the XPS spectrum (Figure S10b in Supporting Information).

4. Fig. 4d, for the Tafel plots, the authors need to explain why they only use the top part to calculate the slopes.

ANS: We thank the referee for his/her comment. We have selected a similar overpotential range (25 ~ 45 mV, the top part) to calculate the Tafel slope for better comparison.

Reviewer #2:

1. The intercalation of layered MoS_2 with other species is well-documented. It is also well-established that Li_xMoS_2 is a strong reducing agent capable of chemically reducing H_2O to produce H_2 leading to exfoliated MoS_2 . So it is not surprising that $1\text{T}'\text{-Li}_x\text{MoS}_2$ can reduce transition metal ions.

ANS: First of all, the intercalation of zero-valent metals into bulk MoS_2 has **never** been reported in the past 50 years (*please differentiate from adsorption on exfoliated and restacked samples*) to the best of our knowledge. It's known that only small, alkali metal ions (Li^+ , Na^+ or K^+) can be electrochemically intercalated into bulk materials due to the narrow spacing and the strong van der Waals force (Main text, Line 62 - 67). As we claimed here, the redox reaction between chemically expanded Li_xMoS_2 and metal ion precursors is utilized as the driving force for intercalation. Such zero-valent metal intercalation is recently developed in MoO_3 (ACS Nano, 2015, 9, 3226), Bi_2Se_3 (J. Am. Chem. Soc. 2015, 137, 5431) etc, however, there has been no studies of the confinement effect on catalysis.

The referee also overlooked what we have written regarding the objectives in the main text (line 82 – 85), where we have explicitly pointed out our goal is to **quantify** the reducing potential of Li_xMoS_2 and convert it into functional materials **without exfoliation**. None of them has been reported before.

2. Since 2H-MoS_2 is a semiconductor, bulk 2H-MoS_2 should suffer from limited electron transfer. The Pt-MoS_2 shows a low charge transfer resistance which could be attributed to Pt enhanced conductivity. However, TEM data (Fig. 8) only reveal the formation of discrete Pt nanoparticles.

ANS: We thank the referee for his/her comment, it must be remembered that we provide electrical energy to help with charge injection, although only a low voltage is applied. We have supplied the TEM images of exfoliated Pt-MoS_2 (Fig. 2), in which Pt nanoparticles are homogeneously dispersed on the MoS_2 nanosheets. Pt-coated MoS_2 has a much lower charge transfer resistance compared to bulk 2H MoS_2 (Fig. S14).

3. Solid evidence is absence to confirm that all Pt nanoparticles are exclusively intercalated into MoS_2 .

ANS: We thank the referee for his/her comment. We have supplied new data on 1) cross-section HRTEM & EELS mapping (Fig. 1), 2) TEM of exfoliated Pt-MoS_2 (Fig. 2), 3) 1D/2D GIXRD spectra (Fig. 3), 4) Small angle x-ray scattering (SAXS) of Pt-MoS_2 (Fig. 4) to support the zero-valent intercalation. We have also exfoliated the sheets and show the

presence of uniformly distributed Pt particles using HRTEM and EELS elemental mapping from inner layers. All these provide compelling evidence for the successful intercalation.

Figure 1. (A) Dark field TEM image of Pt-MoS₂. (B) High resolution HAADF-STEM image and (C-E) EELS mapping showing the intercalation of Pt nanoparticles with an average size of ~ 2 nm in between MoS₂ layer (as indicated by black arrows).

Figure 2. Homogeneous dispersion of Pt nanoparticles on exfoliated Pt-MoS₂ nanosheets: TEM images (a-d), SAED (e) and corresponding EDS spectrum (f). Scale bar: a-c. 50 nm; d. 500 nm; e. 5 nm⁻¹.

Few-layer Pt-MoS₂ was “scotch-tape” exfoliated from single-crystal MoS₂ flake (details in Line 13-16, Page 5 in Supporting Information). Exfoliated nanosheets were then transferred to TEM grids using a PMMA-assisted wet-transfer method. As shown in Figure 2, Pt nanoparticles with sizes ranging from 1.5 to 3.5 nm are homogeneously dispersed on the MoS₂ nanosheets. The selected-area diffraction pattern confirms the typical hexagonal

pattern of single-crystal MoS₂ as well as two continuous rings from {111} and {200} of Pt nanoparticles (as indicated by red-arrow).

Figure 3. Grazing incidence X-ray diffraction (GIXRD) patterns obtained on (a) single crystal MoS₂, (b) Pt-MoS₂ and (c) corresponding 1D GIXRD pattern in the out-of-plane direction. Pt-MoS₂ was prepared using single-crystal MoS₂.

As shown in Figure 3, We can find a broad peak at ~ 2.46 degree (~ 3.6 nm) in 1D and 2D GIXRD for this unique MoS₂ | Pt | MoS₂ sandwiched structure, which is absent in the control, bulk sample. The *d*-spacing of MoS₂ layer is calculated as 0.63 nm from XRD and the average size of Pt nanoparticles is found to be ~ 2 nm from HRTEM images. In such case, Pt-MoS₂ has a repeat unit made of 2 ~ 3 layers of MoS₂ and 1 layer of Pt nanoparticles.

Additionally, we also observed a peak shift for the MoS₂ {002} peak at 14.10 degree. After Pt intercalation, this peak is shifted to 14.34 degree, which can be attributed to the stress from Pt nanoparticles in-between the MoS₂.

Figure 4. Small-angle X-ray scattering (SAXS) profiles of MoS₂ and Pt-MoS₂. The products of the scattering intensity, *I*, and the inverse of form factor of a flat, thin sheet (q^2), $I \cdot q^2$, were plotted against the scattering vector modulus, *q*. The corresponding *d*-spacing values were calculated using the formula $d = 2\pi/q$. Pt-MoS₂ was prepared using single-crystal MoS₂. The SAXS profiles confirm a loosely stacked structure of Pt-MoS₂ due to the hydrogen-bubbles induced volume expansion during zero-valent intercalation.

3.The authors argued that the inner surface is accessible to proton for hydrogen evolution. If it is the case, the high pressure with the accumulated hydrogen bubble in the inner space is

believed to drive the exfoliation of bulk MoS₂. It is supposed that that only Pt species at the edge contributes to the HER activity.

ANS: Actually in the main text, lines 288 – 290, we have explicitly mentioned hydrated salts were used to facilitate the ion-exchange process during intercalation. In such case, bubble-releasing channels are present through cracks, voids, edges, in Pt-MoS₂ to allow pressure release during hydrogen evolution. Additionally, the inner surface of Pt particles is accessible for relatively large, hydrated cupric ions in Fig. 6 in the main text, which indicates the release of H₂ bubbles should not be a problem for Pt-MoS₂. A recent study (Energy Environ. Sci., 2014, 7, 1919) on carbon nanotubes has also shown the enhanced activity and stability towards HER after encapsulation.

4.The use of Pt wire as the counter is also problematic because trace Pt from anode dissolution can enhance the HER activity of the cathodic catalyst.

ANS: Thank you for the comment. HER measurements using **carbon counter** have been carried out and is presented in Figure S19 in the revised version of the paper. There's no major difference in HER activity and stability using carbon counter or platinum counter.

Reviewer #3:

1. The main conclusion the authors would like to deliver is that the small metal nanoparticles (e.g., Pt nanocrystals) are formed in the interlayer gaps of the MoS₂ crystals. However the conclusion is NOT supported with any direct and convincing evidence even though various characterization techniques have been used to characterize the samples.

ANS: We thank the referee for his/her comment. We have supplied new data on 1) cross-section TEM & EELS mapping (Fig. 1), 2) TEM of exfoliated Pt-MoS₂ (Fig. 2), 3) 1D/2D GIXRD spectra (Fig. 3), 4) Small angle x-ray scattering (SAXS) of Pt-MoS₂ (Fig. 4) to support our claims of zero-valent intercalation (See our response to Reviewer #2, question 3). We believe that all these data provide compelling evidence for the successful intercalation. We have exfoliated the sheets and show the presence of uniformly distributed Pt particles using HRTEM and EELS elemental mapping.

2. If the Pt nanoparticles with several nanoparticles are sandwiched in the interlayer gaps of MoS₂, the interlayer spacing the MoS₂ containing Pt nanoparticles have to be on the length scale. However the XRD pattern of the MoS₂ with Pt nanoparticles still show the interlayer spacing of MoS₂ is similar to that of 2H bulk MoS₂ and 1T-Li_xMoS₂ (Fig. 3D showing their XRD peaks at similar positions). This is the first reason that I do not believe that the Pt nanoparticles are confined in small interlayer gaps. The Pt nanoparticles are more likely deposited on the outside of basal surfaces of MoS₂ powders. Otherwise, how can the 6.2-Å gaps accommodate the Pt nanoparticles with several nanoparticles?

ANS: Thank you for the comment. In the case of Pt-MoS₂, the {002} peak of MoS₂ has been rapidly reduced compared to bulk materials and the presence of Pt nanoparticles can be judged clearly from XRD peaks (Main text line 184 - 190). However, as commented by Referee #3, the XRD pattern from 10 to 80 degrees cannot distinguish whether Pt particles are inside or outside. This is because the MoS₂ | Pt | MoS₂ sandwiched structure has a repeat unit of ~ 4 nm, corresponding to a 2 theta angle of ~ 2.5 degree that is beyond the detection limit of conventional XRD.

In this regard, we have conducted 1D/2D grazing incidence XRD (**GIXRD**, see Fig. 3 above) as well as small angle X-ray scattering (**SAXS**, see Fig. 4 above) to prove the intercalation of Pt particles in between MoS₂ layers. As shown in Figure 3, we can find a broad peak at ~ 2.46 degree (~ 3.6 nm) in 1D and 2D GIXRD for this unique sandwiched structure, which is absent in the control, bulk sample. The SAXS profiles also confirm a loosely stacked structure of Pt-MoS₂. These data have been provided in the revised version of the paper.

3. If the Pt nanoparticles are confined in the interlayer gaps of MoS₂, all the single-layer MoS₂ layers should be covered with Pt nanoparticles if the Pt-modified MoS₂ is exfoliated. Can the authors provide the results?

ANS: We thank the reviewer for his/her suggestion. Indeed, we have carried out the **exfoliation** of Pt intercalated single-crystal MoS₂ using "**scotch tape**" method for the

purpose of observing the Pt nanoparticles using TEM *via* a PMMA-based wet-transfer technique. We can observe Pt particles uniformly distributed on MoS₂ flakes with the confirmation of Pt {111}, {200} in SAED pattern and Pt signal in EDS spectrum (Fig. 2). This provides compelling evidence for the successful intercalation of Pt in MoS₂.

4. The ToF-SIMS measurements shown in Fig. 2 canNOT support the conclusion of confining Pt nanoparticles in the interlayer gaps of MoS₂. The scale bars shown in Fig. 2 represent 20 microns, several orders (3-4) larger than the size of Pt nanoparticles as well as the interlayer spacing of MoS₂. How can the authors conclude that the Pt nanoparticles with several nanoparticles are confined in the interlayer gaps of MoS₂. The spatial resolution of the ToF-SIMS mapping cannot support the claim on several-nanometer Pt nanoparticles.

ANS: It must be mentioned that we used single crystal MoS₂ for SIMS measurement (line 141). If the Pt nanoparticles were not intercalated and remained outside, we could only find Pt signal at the surface layer and exposed edges during SIMS measurement. However, as presented in the 3D mapping (Figure 2 in the main text), the Pt nanoparticles are homogeneously distributed in MoS₂ at depths up to 400 nm and beyond. Even though the scale bars (x, y direction) in Fig. 2 are 20 microns, SIMS is known to be very sensitive to **z direction** (length scale at several nanometers). This homogeneous distribution strongly indicates the Pt intercalation in MoS₂.

To address reviewer #3's concern, we have also acquired cross-section HRTEM image and EELS mapping (Fig. 1) to further confirm the Pt intercalation at nanometer scale. The GIXRD data (Fig. 3) also suggests the successful intercalation with a d-spacing of 3.6 nm.

5. It is questionable of the results shown in Fig. S11 (Mohr Titration of chloride ions). The chloride ions existing in reaction solution can be significantly contributed by the Pt precursor ions (*i.e.*, PtCl₆²⁻). As a result, the measured concentration of Cl⁻ is not solely determined by the exchanged Cl⁻ from the 1T-LixMoS₂ sample. Therefore, the conclusion derived from this measurement is not reasonable.

ANS: Mohr titration of Cl⁻ ions was used to determine the amount of residual chloride ions after *in-situ* reduction of Pt precursor. The amount of PtCl₆²⁻ was carefully controlled to ensure complete reaction (calculated from weight% of Na₂PtCl₆ verse the actual Pt loading in PtMoS₂, >95 % Pt is found in Pt-MoS₂ based on elemental analysis). Since PtCl₆²⁻ has completely reacted with Li_xMoS₂ (Fig. S4, see color change from yellow to clear in the supernatant), most chloride ions have **intercalated** into MoS₂ layers and the residual Cl⁻ ions in the solution are negligible.

The reliability of Mohr titration is quite good as presented in the working curve in Fig. S11. Therefore, Li⁺ ions are mostly ion-exchanged from MoS₂ host together with Cl⁻ counter-ions after zero valent intercalation.

Yours sincerely

Kian Ping Loh

Professor

Department of Chemistry

National University of Singapore

Email: chmlhkp@nus.edu.sg

Reviewers' Comments:

Reviewer #1 (Remarks to the Author)

The authors did additional experiments to show the coverage of Pt increase with time. Nevertheless, they have not explained why the Pt concentration has random peak regions. The TEM image in Fig. S6 seems to only indicate that Pt is located in the edges.

The authors attribute the poor stability of Pd-MoS₂ to the presence of PdO. PdO may lead to worse HER activity, but it is not clear why PdO will lead to poor stability since PdO was formed during the intercalation step.

Reviewer #2 (Remarks to the Author)

It seems that the authors have addressed most concerns raised by reviewers and acceptance is recommended.

Reviewer #4 (Remarks to the Author)

The main conclusion the authors would like to deliver is that the small metal nanoparticles (e.g., Pt nanocrystals) are formed in the interlayer gaps of the MoS₂ crystals. The authors provide various characterizations such as TEM, EELS, SAXS, GIXRD, GIXRD and SIMS to prove this. All these data together proved it is possible to get Pt nanoparticles intercalated MoS₂, and, also, addressed the concerns of reviewer 3. However, the 'intercalated' used in the title 'Interface Confined Hydrogen Evolution Reaction in Zero Valent Metal-Intercalated Molybdenum Disulfide' is inappropriate. Usually, people use intercalation to describe atoms, small molecule or ions rather than nanoparticles. Or 'zero valent metal nanoparticles-intercalated MoS₂' should be used here to reduce the ambiguity.

Response to Referees:

Reviewer #1:

We thank the Reviewer for his/her positive assessment. To answer his concern:

1. *“The authors did additional experiments to show the coverage of Pt increase with time. Nevertheless, they have not explained why the Pt concentration has random peak regions. The TEM image in Fig. S6 seems to only indicate that Pt is located in the edges.”*

We thank the referee for his/her comment. As shown in Figure 3D and E, the Pt nanoparticles are homogeneously distributed on the MoS₂ nanosheets in the exfoliated single-crystal samples. This provides strong evidence that Pt nanoparticles are uniformly distributed and are not only accumulated at the edge regions of MoS₂. It must be noted that Fig. S6 is a **cross-section TEM** image. Due to its low TEM contrast, we are only able to image those nanoparticles at the edges and not possible to see the particles situated in the inner regions (especially for those located on the central plane). In other words, the purpose of this cross-section TEM image is only to prove the successful intercalation in between MoS₂ layers (as indicated by black arrows).

To address the reviewer's concern, we have revised the manuscript and SI as follow:

Main text, Page 8

“As shown in Figure 3D, 3E, Pt nanoparticles with sizes ranging from 1.5 to 3.5 nm are homogeneously dispersed on the MoS₂ flakes.”

Supplementary Info.

“EELS mapping showing the intercalation of Pt nanoparticles with an average size of ~ 2 nm in between MoS₂ layers (as indicated by black arrows). Since this is a cross-section image, we are only able to image those nanoparticles at the edges, whereas intercalated nanoparticles (especially for those located on the central plane) in the inner regions cannot be imaged due to the low TEM contrast.”

2. *“The authors attribute the poor stability of Pd-MoS₂ to the presence of PdO. PdO may lead to worse HER activity, but it is not clear why PdO will lead to poor stability since PdO was formed during the intercalation step.”*

We thank the referee for his/her comment. PdO can lead to worse HER activity as well as poor stability. This is because PdO has a much **higher solubility in acidic condition** (0.5 M sulfuric acid) compared to zero-valent Pd particles. The oxidation tendency of Pd(0) is also much higher than Pt(0). Once the surface oxide is dissolved, fresh surface of Pd particles will be exposed to H₂O or trace amount of O₂ etc, leading to further oxidation and degradation during long-term operation. Such dissolution and redeposition has been regarded as one major degradation mechanisms for noble metal catalysts (See supplementary note 3, Pt as an example).

To address the reviewer's concern, we have revised the manuscript and SI as follow:

Main text, Page 12

“while the relatively poor stability of Pd-MoS₂ may be attributed to its surface oxidation and dissolution of its oxide in the acidic environment.”

Supplementary Note 5. Discussion on HER stability:

“The relatively poor stability of Pd-MoS₂ can be ascribed to the partial oxidation of Pd nanoparticles as evidenced by XPS spectra in Supplementary Figure 9B (21) since PdO has a much higher solubility in acidic condition.”

Reviewer #2:

We thank the Reviewer for his/her positive assessment.

“It seems that the authors have addressed most concerns raised by reviewers and acceptance is recommended.”

Reviewer #4:

We thank the Reviewer for his/her positive assessment. To answer his concern:

“The main conclusion the authors would like to deliver is that the small metal nanoparticles (e.g., Pt nanocrystals) are formed in the interlayer gaps of the MoS₂ crystals. The authors provide various characterizations such as TEM, EELS, SAXS, GIXRD, GIXRD and SIMS to prove this. All these data together proved it is possible to get Pt nanoparticles intercalated MoS₂, and, also, addressed the concerns of reviewer 3.”

1. “However, the ‘intercalated’ used in the title ‘Interface Confined Hydrogen Evolution Reaction in Zero Valent Metal-Intercalated Molybdenum Disulfide’ is inappropriate. Usually, people use intercalation to describe atoms, small molecule or ions rather than nanoparticles. Or ‘zero valent metal nanoparticles-intercalated MoS₂’ should be used here to reduce the ambiguity.”

We thank the reviewer for his/her suggestion. According to his suggestion, we have changed our title to

“Interface Confined Hydrogen Evolution Reaction in Zero Valent Metal Nanoparticles-Intercalated Molybdenum Disulfide”

Yours sincerely

Kian Ping Loh

Professor

Department of Chemistry

National University of Singapore

Email: chmlhkp@nus.edu.sg